# Estimating the prevalence of discrepancies between study registrations and publications: a systematic review and meta-analyses

TARG Meta-Research Group & Collaborators

**Correspondence to**
Robert Thibault;
robert.thibault@stanford.edu

## ABSTRACT

**Objectives** Prospectively registering study plans in a permanent time-stamped and publicly accessible document is becoming more common across disciplines and aims to reduce risk of bias and make risk of bias transparent. Selective reporting persists, however, when researchers deviate from their registered plans without disclosure. This systematic review aimed to estimate the prevalence of undisclosed discrepancies between prospectively registered study plans and their associated publication. We further aimed to identify the research disciplines where these discrepancies have been observed, whether interventions to reduce discrepancies have been conducted, and gaps in the literature.

**Design** Systematic review and meta-analyses.

**Data sources** Scopus and Web of Knowledge, published up to 15 December 2019.

**Eligibility criteria** Articles that included quantitative data about discrepancies between registrations or study protocols and their associated publications.

**Data extraction and synthesis** Each included article was independently coded by two reviewers using a coding form designed for this review (osf.io/728ys). We used random-effects meta-analyses to synthesise the results.

**Results** We reviewed k=89 articles, which included k=70 that reported on primary outcome discrepancies from n=6314 studies and, k=22 that reported on secondary outcome discrepancies from n=1436 studies. Meta-analyses indicated that between 29% and 37% (95% CI) of studies contained at least one primary outcome discrepancy and between 50% and 75% (95% CI) contained at least one secondary outcome discrepancy. Almost all articles assessed clinical literature, and there was considerable heterogeneity. We identified only one article that attempted to correct discrepancies.

**Conclusions** Many articles did not include information on whether discrepancies were disclosed, which version of a registration they compared publications to and whether the registration was prospective. Thus, our estimates represent discrepancies broadly, rather than our target of *undisclosed* discrepancies between *prospectively* registered study plans and their associated publications. Discrepancies are common and reduce the trustworthiness of medical research. Interventions to reduce discrepancies could prove valuable.

**Registration** osf.io/ktmdg. Protocol amendments are listed in online supplemental material A.

### STRENGTHS AND LIMITATIONS OF THIS STUDY

⇒ We employ a wide-reaching search strategy and captured 89 articles including over 6000 registrations and publications.

⇒ Our coding procedure includes fine-grained information that allows us to run meta-regressions and test whether several parameters impact discrepancies.

⇒ All our data and code are openly available.

⇒ The high heterogeneity in the meta-analyses led to wide-ranging CIs and prediction intervals.

⇒ Many articles did not fully operationalise their definition of what constitutes a discrepancy (eg, which version of the registration was used).

## INTRODUCTION

In 2000, ClinicalTrials.gov and the ISRCTN Registry were launched with several aims, including aiding participant recruitment, facilitating knowledge synthesis and reducing duplication, publication bias and selective reporting.[1] In 2005, the International Committee of Medical Journal Editors (ICMJE) made *prospective* registration a condition of consideration for publication.[2] Thousands of journals now claim to follow this policy.[3] In parallel, the WHO International Clinical Trials Registry Platform established a minimum set of required information for a trial to be considered fully registered, including experimental design elements such as the conditions being studied, intervention, key inclusion and exclusion criteria, sample size, primary outcomes and key secondary outcomes.[4] While the relatively widespread uptake of clinical trial registration has substantially improved transparency, many trials remain unregistered, are registered after enrolment of participants begins or analyses are complete (ie, *retrospective* registration), are never published, or publish outcomes discrepant with those in the registration without disclosing the discrepancy.[5 6]

Nevertheless, the existence of registries allows researchers to identify and quantify these issues.

In research disciplines other than clinical trials, study registration is becoming more common, but remains far from standard practice.[7–10] For example, starting around 2011 the field of psychology has increasingly taken the 'replication crisis' seriously and many researchers and journals now use registration to reduce bias and make risk of bias transparent. Other disciplines have created dedicated registries, such as PROSPERO for systematic reviews and the American Economic Association's registry for randomized controlled trials (AEA RCT Registry).

In this manuscript, we systematically reviewed articles that quantify the prevalence of discrepancies between registrations or study protocols and their associated publications (eg, in primary outcome measures). Our analysis extended beyond the three systematic reviews already published on this topic in several ways.[11–13] First, registration has expanded beyond clinical trials; we included all research disciplines and used key word searches for registries including the Open Science Framework, the AEA RCT Registry and PROSPERO. Second, we extracted more fine-grained information about a wide range of discrepancies (eg, outcomes, analysis, sample size), as well as which version of the registration was surveyed and whether discrepancies were disclosed (we believe disclosed discrepancies present little reason for concern). Third, our review included over twice as many studies as previous systematic reviews on this topic, provided meta-analytical estimates and used meta-regression and additional analyses to attempt to identify predictors of discrepancies.

## METHODS

### Terminology

We present a systematic review of k=89[5 6 14–101] articles that assessed a wide range of outcome discrepancies and non-outcome discrepancies across over n=7000 studies. To avoid confusion, this report consistently uses the terms *studies* to refer to the over n=7000 individual studies that were *assessed*, and the term *article* to refer to the k=89 articles that assessed these studies, and that we *reviewed*. We restrict our usage of the term *publication* to refer to the publications stemming from the *studies* (not to refer to the *articles*).

We use the term *discrepancy* to refer to any incongruity between the content of a publication and its associated registration (eg, on ClinicalTrials.gov) or study protocol (eg, submitted to an ethics review board or funding agency)—see box 1, for examples. We use the term *prospective registration* broadly to include terms used in different research disciplines, such as prospective trial registration, preregistration and pre-analysis plans. All these terms indicate the registration of study details *before* commencing a study, or in some cases, before viewing the data or removing the blind. They are in contrast to *retrospective* registration, which occurs after participant

---

> **Box 1  Examples of discrepancies**
>
> We coded 10 types of outcome discrepancies and 10 types of non-outcome discrepancies . The degree to which the information in a registration is discrepant with the information in a publication can range widely. The associated concern about risk of bias can also range widely and often requires domain expertise to assess. We present two examples verbatim from the study by Calméjane *et al*[98] in this box. Many more examples are available in the appendix of Calméjane *et al*[98] and at https://www.compare-trials.org/results. Some researchers suggest that people checking for discrepancies should compare publications to clinical trial study protocols instead of registrations, because they are more thorough and more likely to be updated when trialists change their plans.[114]
>
> **Registered:** Primary outcome: Visual acuity (time frame: 6 months).
>
> **Published:** Primary outcome: 3-year cumulative incidence rate of myopia. Myopia was defined as a spherical equivalent refractive error (sphere+½ cylinder) of at least −0.50 D.
>
> **Coded as:** Timing of outcome measurement changed.
>
> **Registered:** Primary outcome: Severity of device and procedure-related complications (time frame: At the time of ExAblate Transcranial thalamotomy procedure). Secondary outcome: Effectiveness of the ExAblate Transcranial MRgFUS treatment determined using the Clinical Rating Scale for Tremor (CRST) (time frame: Participants will be followed from the date of treatment until study completion, approximately up to 12 months).
>
> **Published:** Primary outcome: Change from baseline to 3 months in the tremor score for the hand derived from the CRST, Part A (three items: resting, postural and action or intention components of hand tremor), and the CRST, Part B (five tasks involving handwriting, drawing and pouring).
>
> **Coded as:** Secondary outcome promoted to primary outcome; timing of outcome measurement changed; primary outcome omitted.

---

enrolment begins or analyses are complete. We use the term *outcome discrepancy* to indicate a discrepancy in the outcome measure registered versus the outcome measure reported in a publication (not to indicate a discrepancy in the value of a reported outcome between these documents).

### Eligibility criteria

We included articles that reported quantitative data about discrepancies between registrations or study protocols and their associated publication. We excluded conference proceedings and articles written in a language other than English (for full inclusion and exclusion criteria, see our preregistered protocol at osf.io/ktmdg).

### Study selection

We searched Scopus and Web of Science on 15 December 2019 using the queries in online supplemental appendices A and B of our preregistered protocol (osf.io/ktmdg). Briefly, our queries included (1) variations of the terms preregistration, pre-analysis plans and prospective registration in the title or keyword fields; (2) terms indicating discrepancies such as 'outcome switching' in the title, keywords or abstract; (3) names of registration or protocol repositories such as 'clinicaltrials.gov' in the title or keywords; and excluded overlapping but

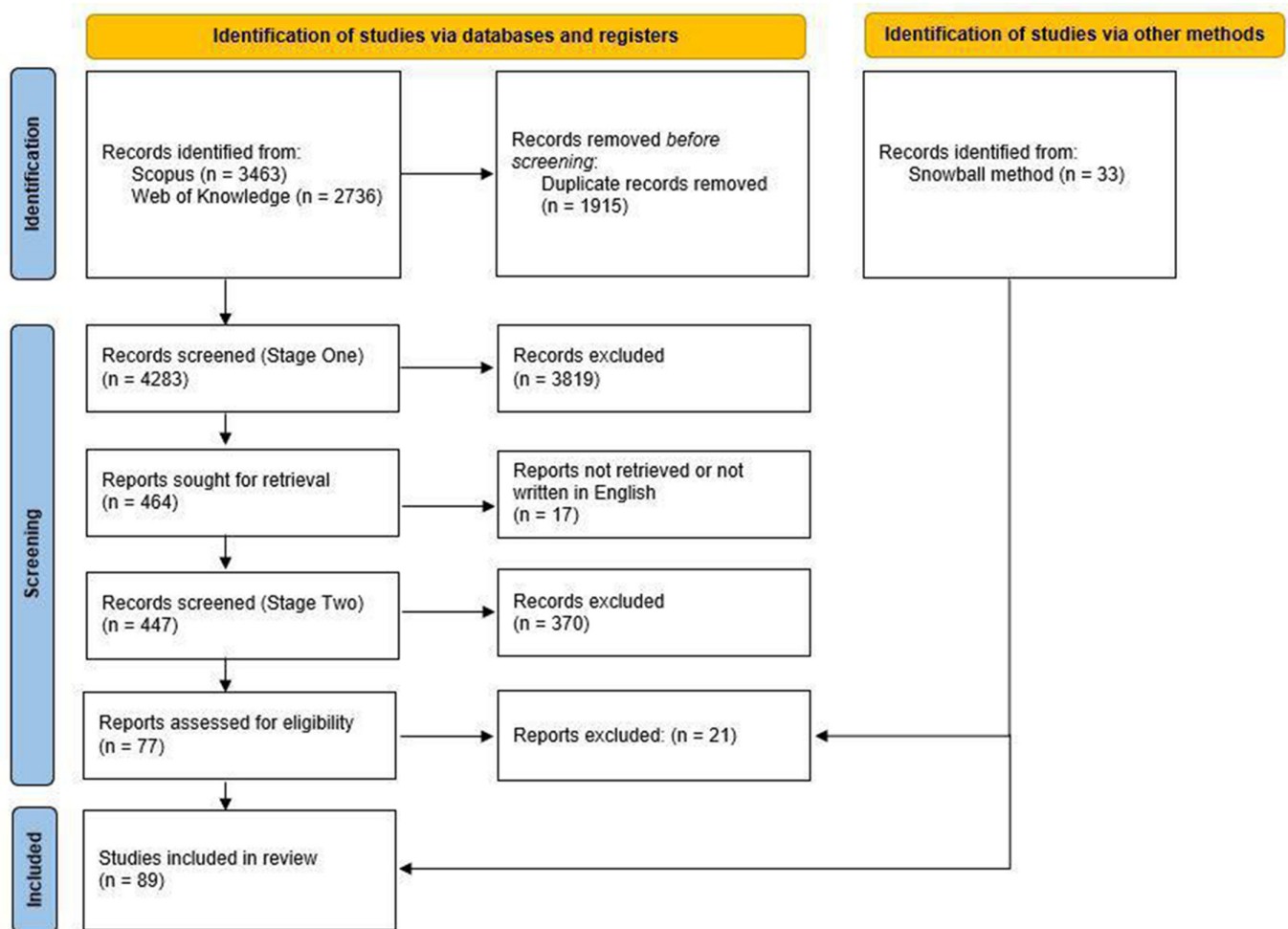

**Figure 1** Preferred Reporting Items for Systematic Reviews and Meta-Analyses (PRISMA) flowchart of article inclusion.

irrelevant terms (eg, 'nursing preregistration'). To limit the number of irrelevant articles, we did not search for variations of the term preregistration or for repository names in the abstract field.

Our search returned 4283 articles after duplicates were removed (see figure 1 for a PRISMA flowchart). Articles were screened independently by two reviewers in two stages. In Stage One, reviewers screened titles and, if necessary, briefly examined abstracts of articles to determine inclusion in the systematic review or in a scoping review (details at https://osf.io/ktmdg) on prospective registration. If at least one of the reviewers deemed an article potentially relevant, it was included in Stage Two screening. In Stage Two, the reviewers independently examined the remaining 464 abstracts in greater detail for eligibility. Disagreements were resolved through discussion between the two reviewers and eventual consensus. Inter-rater agreement for the 464 articles was Cohen's k=0.67 for inclusion in the systematic review (the list of articles and coding is available at osf.io/wa62f). Inter-rater agreement for all 4283 articles was Cohen's k=0.72. We then used a snowball method and identified 33 additional articles that met our inclusion criteria, mostly through citations in studies by Li *et al*[11] and Jones *et al*[13]. These 33 additional articles are not included in the

inter-rater agreement scores. After a full-text review, we included 89 articles in our systematic review.

## Coding items

Each included article was independently coded by two of four reviewers (RTT, RC, OvdA and SW) using a coding form designed for this review. The form consisted of five sections that assessed (1) article characteristics, (2) study registration details, (3) 10 types of outcome discrepancies, (4) 10 types of non-outcome discrepancies and (5) any additional descriptive or inferential statistics on discrepancies. The form details the operationalisation of each variable we coded, and is available at osf.io/728ys. We chose items to code based on a pilot test of our protocol, as well as the categories used in a seminal paper[94] and a systematic review on discrepancies.[11]

The data we extracted was often presented as summary results in a table and sometimes as text in the results section. To be included in our meta-analyses this data had to include at least two of (1) the number of studies assessed (denominator), (2) the number of studies with a given discrepancy (numerator) and (3) the percentage of studies with a given discrepancy—from which we could calculate an unreported numerator or denominator. We did not access the raw data. If an article did not report

data for a certain measure (eg, secondary outcome discrepancies), then we did not include that article in the meta-analysis for that measure (this is why k varies among the meta-analyses we present). For the meta-regressions we performed, we could not find data on (1) the version of the registry that publications were compared with for 35 articles, and (2) the number of studies that disclosed discrepancies for 57 articles. We coded these cases as 'not reported' and included 'not reported' as a factor in the meta-regressions. All other meta-regression data was complete. Further coding details are available in online supplemental material B.

The complete data set, including the coding of each reviewer and the resolved coding, is available at osf.io/ ue2c6. A cleaned data set with only the resolved coding is available at osf.io/6cn9m.

### Statistical analyses

We performed two main random-effects meta-analyses: one on the proportion of studies with at least one primary outcome discrepancy, and another on the proportion of studies with at least one secondary outcome discrepancy. We used random-effects models because they allow for the true effect to vary across the populations the articles sampled from, and the articles we reviewed differ in their methodologies and the research disciplines that they assess. We used a random intercept logistic regression model with the Knapp-Hartung adjustment for the synthesis of proportions.[102] We used the maximum-likelihood method for estimating the between-study heterogeneity (tau). We also performed meta-regressions to test whether article characteristics are associated with the proportion of studies with at least one primary or secondary outcome discrepancy.

For pooled estimates, we report both CIs and prediction intervals. Whereas researchers are likely more familiar with CIs, interpreting CIs can be unintuitive,[103] and their pooled-estimate does not incorporate uncertainty due to the between-article heterogeneity. If we assume that we could resample from our population, 95% of the resampled *meta-analyses* would yield a 95% CI that contains the true value of the parameter being estimated (eg, proportion of articles with at least one primary outcome discrepancy). Alternatively, if we are interested in the results that would come from another *article* assessing discrepancies, we would want a 95% prediction interval. In other words, of 100 articles drawn from the same population, we could expect the results from 95 of them—on average—to fall within the 95% prediction interval. While prediction intervals are not commonly reported, methodologists recommend reporting them for random-effects meta-analysis, particularly when few articles are included or, as in our case, included articles are highly heterogeneous.[104 105]

Whereas we did not perform a formal risk of bias assessment—because our review differed substantially from the purpose these tools were built for—we shed light on a few potential sources of bias with additional analyses that consider the funding source, statistical significance and the timing of registration of included *studies*. These additional analyses were not prospectively registered. We made a few amendments to our preregistered study protocol which are listed in online supplemental material A.

### Patient and public involvement

Patients and members of the public were not involved in the design, conduct or reporting of this systematic review.

## RESULTS

### Articles characteristics

We identified and reviewed k=89 articles that report at least one type of discrepancy. Articles that checked for outcome discrepancies assessed a median of 68 studies (IQR: 33–112). Article characteristics are outlined in table 1. All articles except for two, one preprint in economics[95] and one preprint in psychology,[96] focused on clinical trials or systematic reviews. All but k=10 articles were solely observational. Only one article attempted to correct published discrepancies.[97] The authors of this article assessed all trials published in five journals over a 6-week period and sent a letter to the editor for each trial that published a discrepant outcome (for 58 letters in total). In the time since our literature search, at least two more interventional studies were published. One reports a trial that attempted to reduce discrepancies at medical journals by sending peer reviewers information about the study registration.[106] They found null results. The other was a feasibility study that assigned a peer reviewer to specifically check for discrepancies in manuscripts submitted for publication.[107] Further details about article characteristics are available in online supplemental material C.

### Registration timing

Articles varied in the level of detail they provided about whether and when studies were registered. For example, whereas some articles presented their sample only after selecting for prospectively registered studies, other articles detailed their selection process including how many studies were registered and if so, when they were registered. Using the terminology in the articles we reviewed, articles identified studies that were registered retrospectively (k=29), registered during participant enrolment (k=17), registered after participant enrolment was complete (k=14) and studies that were not registered (k=36). Several articles identified non-registered studies in their sampling process, but did not include these studies in their final sample. Many articles were ambiguous regarding when some studies were registered (k=47) and whether or not some studies were registered at all (k=24). While these data do not provide fine-grained detail, they highlight two overarching issues: many studies are not registered, and many registered studies are registered retrospectively. These studies fail to meet the Declaration

**Table 1** Article characteristics

| Article characteristic | k=89 | n=6929 |
|---|---|---|
| Discipline | | |
| Medicine | 81 | 6452 |
| Dentistry | 3 | 254 |
| Psychology | 3 | 68 |
| Economics | 1 | 93 |
| Physical therapy | 1 | 62 |
| Source of registration or protocol assessed for discrepancies | | |
| Registry | 73 | 6107 |
| Ethics application | 7 | 146 |
| Other protocol | 5 | 466 |
| Marketing application | 2 | 126 |
| Grant application | 2 | 84 |
| Sources searched to identify studies | | |
| Journals | 32 | 3264 |
| Registries | 23 | 1547 |
| Search engines | 16 | 1527 |
| Ethics boards | 7 | 146 |
| Funders | 3 | 96 |
| Registries and search engines | 3 | 27 |
| Regulators | 2 | 126 |
| Research group | 2 | 44 |
| Registries and journals | 1 | 152 |
| Type of study | | |
| Solely observational | 79 | 6344 |
| Observational and study authors were contacted | 9 | 518 |
| Observational and interventional | 1 | 67 |
| Version of registry that publications were compared with | | |
| Original | 29 | 2607 |
| Most recent | 15 | 1281 |
| Other version/unclear | 10 | 670 |
| Not reported | 35 | 2371 |
| Number of studies within each article that disclosed discrepancies | | |
| One or more | 19 | 1547 |
| None | 9 | 441 |
| Article excluded publications with disclosed discrepancies | 4 | 326 |
| Not reported | 57 | 4615 |

of Helsinki[108] (item 35) requirement that 'Every research study involving human subjects must be registered in a publicly accessible database before recruitment of the first subject' and the equivalent ICMJE policy,[109] which thousands of journals claim to follow.[3]

Eighty of the k=89 articles we reviewed report at least one type of outcome discrepancy. Of these, 23 report only on studies that were unambiguously prospectively registered, 51 do not unambiguously distinguish between prospectively and retrospectively registered studies and 6 report outcome discrepancies separately for each of prospectively and retrospectively registered studies. Separate meta-analyses for unambiguously prospectively registered studies and studies with unclear timing of registration are presented in online supplemental material D.

Forty-six of the k=89 articles report at least one non-outcome discrepancy (eg, in sample size or analyses). Of these, 12 report only on studies that were unambiguously prospectively registered, 33 do not unambiguously distinguish between prospectively and retrospectively registered studies and 1 reports non-outcome discrepancies separately for each of prospectively and retrospectively registered studies.

### Primary outcome discrepancies

An estimated 29% to 37% (95% CI) of the population of studies contained at least one primary outcome discrepancy (figure 2). The 95% prediction interval is 10% to 68%.

This meta-analysis had high heterogeneity ($I^2$=86%), suggesting that the broad range of estimates across the articles stem largely from differences in the methodology of the articles or populations they sample from, rather than from chance. Heterogeneity could not be explained by meta-regression of any of the following article-level characteristics: discipline (p=0.28), whether the publications were compared with registry entries versus other protocol formats (eg, ethics applications) (p=0.46), sources searched to identify studies (p=0.65), version of the registry analysed (p=0.77), whether discrepancies were disclosed (p=0.97) and year of article publication (p=0.83). The meta-regression on discipline had low power because 63 articles assessed medical research and 7 assessed studies across dentistry, psychology, physical therapy and economics. To increase statistical power, we reran this meta-regression after dichotomising discipline and found that non-medical disciplines may have a greater proportion of studies with at least one primary outcome discrepancy (p=0.09; OR 95% CI: 0.91 to 3.19). We ran another meta-regression after dichotomising the source which publications were compared with—into registrations versus other protocols—and did not find evidence to suggest this moderator played a role (p=0.42). We also conducted a sensitivity analysis that included all six article-level characteristics in a single meta-regression. We found that publications compared with the most recent version of a registration may have a smaller proportion of studies with at least one primary outcome discrepancy, relative to publications that were compared with the original version of a registration (p=0.08; OR 95% CI: 0.32 to 1.07). All meta-regression model summaries are presented in online supplemental material E.

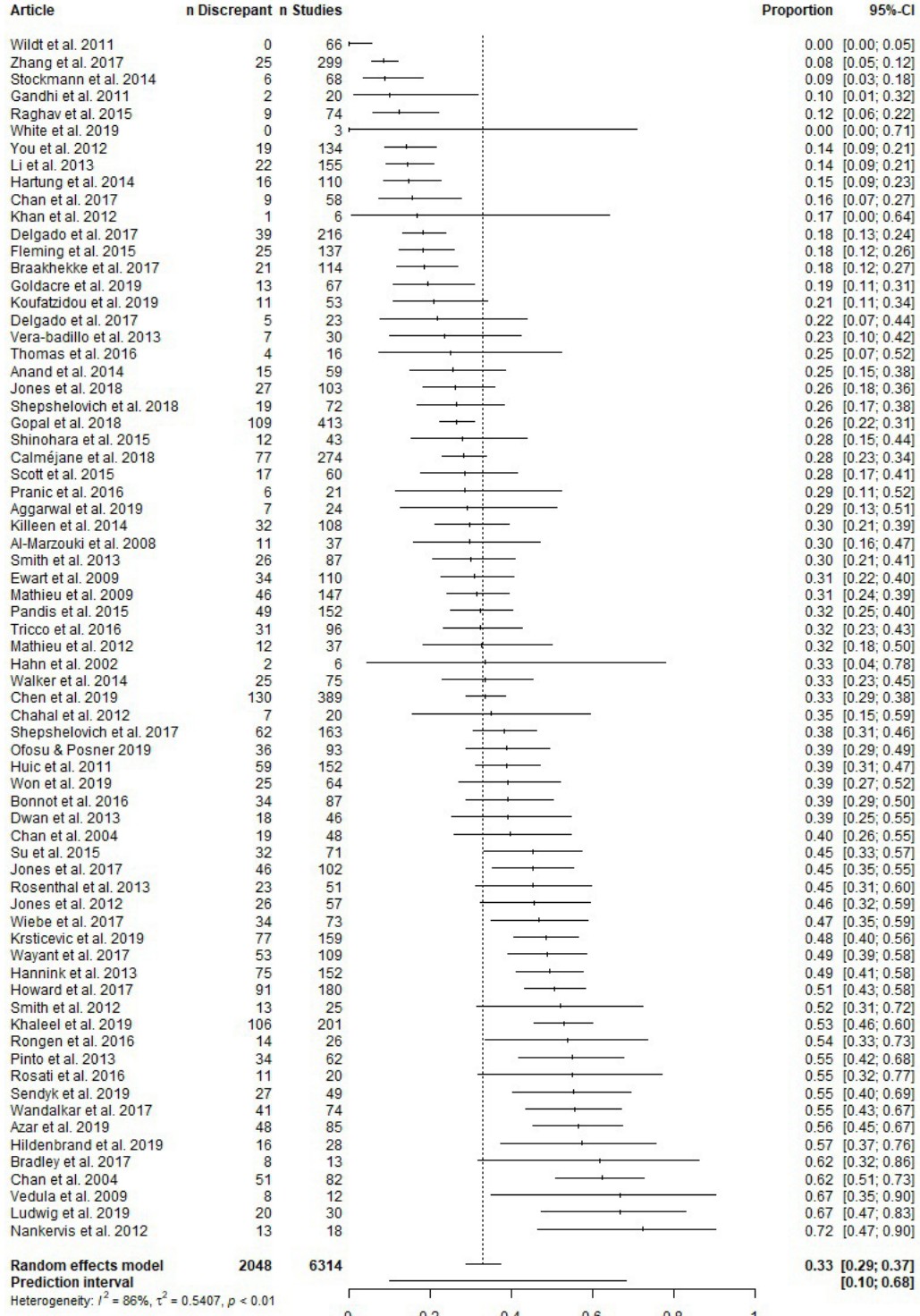

**Figure 2** Forest plot of articles reporting the proportion of assessed studies with at least one primary outcome discrepancy.

The high heterogeneity in this meta-analysis may stem from genuine differences among the articles, including the subdisciplines surveyed, specific sources searched, definition of a discrepancy (eg, whereas some articles considered a change in the timing of an outcome as a discrepancy, others did not) and other article characteristics that may or may not have been reported. Our data set contains more fine-grained information about the specific subdiscipline surveyed and specific sources searched. While we do not further explore these potential moderators in the present report, we note that, whereas some subdisciplines and sources were highly specific (eg, cystic fibrosis, lung cancer immunotherapy, Global Resource of Eczema Trials database), others were broad (eg, medicine, ClinicalTrials.gov, core clinical MEDLINE journals). We did not collect information on the exact definitions

**Table 2** Meta-analytical estimates for the proportion of studies that contain various types of outcome discrepancies

|  | Discrepant studies (95% CI) | Discrepant studies (95% PI) | k | n |
|---|---|---|---|---|
| Any outcome discrepancy | 41 to 75% | 7 to 97% | 18 | 1113 |
| Any primary outcome discrepancy | 29 to 37% | 10 to 68% | 70 | 6314 |
| Any secondary outcome discrepancy | 50 to 75% | 13 to 95% | 22 | 1436 |
| Primary outcome demoted to secondary outcome | 6 to 10% | 2 to 31% | 51 | 4560 |
| Primary outcome omitted | 6 to 12% | 1 to 43% | 51 | 4338 |
| Primary outcome added | 7 to 11% | 2 to 34% | 54 | 4697 |
| Secondary outcome promoted to primary outcome | 4 to 6% | 1 to 19% | 46 | 4135 |
| Secondary outcome omitted | 16 to 35% | 4 to 72% | 18 | 1243 |
| Secondary outcome added | 19 to 43% | 3 to 83% | 20 | 1305 |
| Timing of outcome measurement changed | 6 to 16% | 1 to 62% | 30 | 3472 |

We only coded 'any outcome discrepancy' for articles that checked for both primary and secondary outcome discrepancies in the studies they assessed. Forest plots for all meta-analyses in this table are in online supplemental material F.
PI, prediction interval.

an article used to identify a primary outcome discrepancy. However, we did collect information on the proportion of articles with subcategories of outcome discrepancies, which are more strictly defined and listed in table 2 (eg, promoting a secondary outcome to a primary outcome). We ran meta-analyses on these subcategories of outcome discrepancies and found they also had high heterogeneity (table 2). Thus, varying definitions are unlikely to be the main driver of the high heterogeneity in the present analysis on primary outcome discrepancies.

### Secondary outcome discrepancies

An estimated 50% to 75% (95% CI) of the population of studies contained at least one secondary outcome discrepancy (figure 3). The 95% prediction interval is 13% to 95% .

This meta-analysis also had high heterogeneity (I²=90%) which could not be explained by meta-regression of the version of the registry analysed (p=0.80) or the year of article publication (p=0.72). Meta-regression of the sources searched to identify studies explained some heterogeneity, in that searches stemming from journals, compared with registries, had a greater proportion of publications with at least one secondary outcome discrepancy (p=0.03; OR 95% CI: 1.89 to 13.45). Meta-regressions on discipline (p=0.29), whether discrepancies were disclosed (p=0.68) and whether the publications

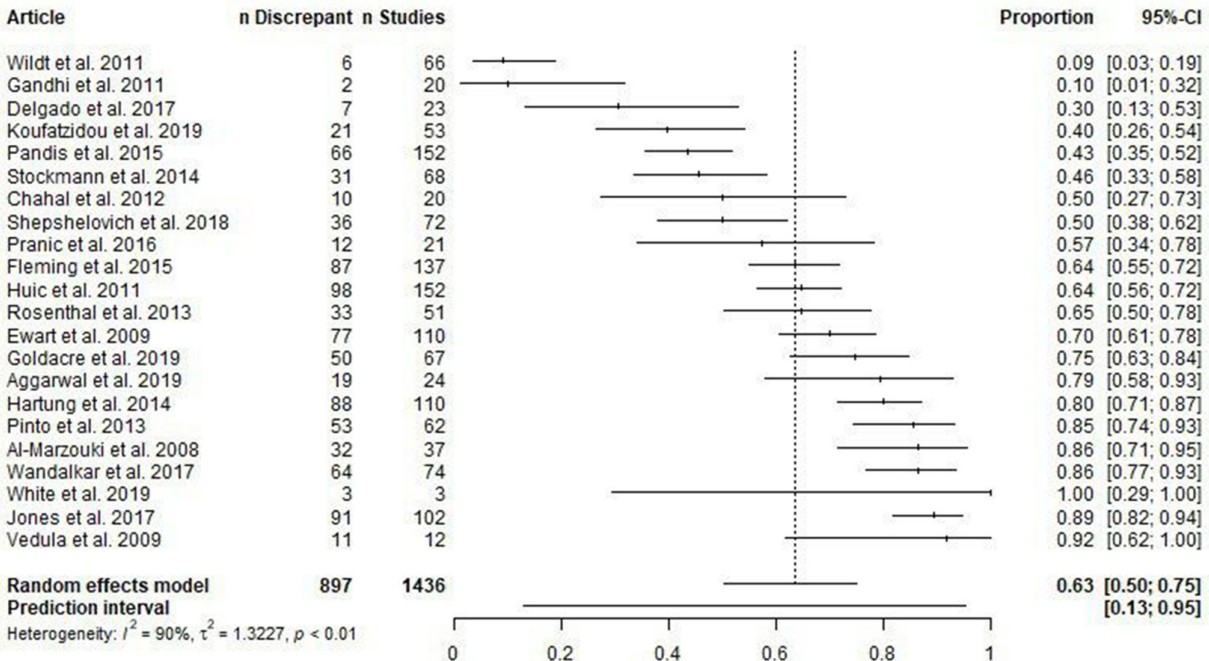

**Figure 3** Forest plot of articles reporting the proportion of assessed studies with at least one secondary outcome discrepancy.

**Table 3** Additional analyses regarding discrepancies

| Analysis | 95% CI | 95% PI | k | n |
|---|---|---|---|---|
| Percentage of studies with at least one outcome discrepancy that disclose an outcome discrepancy | 4 to 19% | 0.3 to 74% | 21 | 620 |
| Percentage of outcome discrepancies that favoured statistically significant results | 49 to 66% | 23 to 86% | 24 | 671 |
| Likelihood ratio of a study with versus without an outcome discrepancy to contain statistically significant results | 0.56 to 1.06 | 0.42 to 1.43 | 7 | 405 |
| Likelihood ratio of a study with versus without a statistically significant results to contain an outcome discrepancy* | 0.64 to 0.99 | 0.59 to 1.06 | 7 | 405 |
| Likelihood ratio of a study with versus without industry funded to contain an outcome discrepancy | 0.61 to 0.91 | 0.44 to 1.27 | 22 | 2623 |
| Likelihood ratio of a prospectively registered study versus retrospectively registered study to contain a primary outcome discrepancy | 0.46 to 2.57 | 0.13 to 8.85 | 4 | 260 |

*This analysis uses the same data as the likelihood ratio analysis before it. One of these seven articles checked discrepancies in analyses, rather than discrepancies in outcome measures. Forest plots for all meta-analyses in this table are in online supplemental material G.
PI, prediction interval.

were compared with registry entries versus other protocol formats (p=0.08) had very low statistical power because almost all articles had the same characteristic. All meta-regression model summaries are included in online supplemental material E.

Descriptively, omitting secondary outcomes and adding secondary outcomes appears to occur more frequently than omitting primary outcomes, adding primary outcomes or demoting primary outcomes, which in turn appear to occur more frequently than promoting a secondary outcome (see table 2).

### Parameters potentially related to outcome discrepancies

A subset of articles contained information on parameters potentially related to the proportion of outcome discrepancies. These include the disclosure of discrepancies, presence of a 'statistically significant' result, funding source and timing of registration (table 3).

### Non-outcome discrepancies

The meta-analyses for the non-outcome discrepancies had high heterogeneity, and wide CIs and prediction intervals (table 4). Articles varied in the criteria they used to identify non-outcome discrepancies and there were fewer articles than for outcome discrepancies. Prediction intervals can be particularly imprecise when few articles are included in a meta-analysis.[110] Whereas our coding procedure divided outcome discrepancies into 10 subcategories, it did not employ the same level of granularity for non-outcome discrepancies. Online supplemental material H contains additional information on non-outcome discrepancies.

### Gaps in the literature

We identified several gaps in the literature on discrepancies. There exists little research on: (1) the prevalence of discrepancies in fields other than clinical research, (2)

**Table 4** Meta-analytical estimates for the proportion of studies that contain various types of non-outcome discrepancies

| | Discrepant studies (95% CI) | Discrepant studies (95% PI) | k | n |
|---|---|---|---|---|
| Eligibility criteria | 25 to 57% | 5 to 90% | 15 | 1153 |
| Sample size | 26 to 44% | 8 to 78% | 25 | 1398 |
| Randomisation | 2 to 64% | 0.06 to 98% | 5 | 176 |
| Blinding | 5 to 42% | 0.04 to 99% | 3 | 224 |
| Intervention | 3 to 52% | 0.1 to 97% | 7 | 550 |
| Study duration | 3 to 89% | 0.02 to 99.94% | 4 | 184 |
| Analysis | 19 to 52% | 4 to 86% | 12 | 404 |
| Subgroup analysis | 35 to 93% | 2 to 99.7% | 9 | 545 |
| Funding | 7 to 84% | 0.2 to 99.5% | 5 | 212 |
| Results | 7 to 82% | 0.2 to 99% | 6 | 262 |

Values less than 0.5% and greater than 99.5% are rounded to one significant digit from 0 to 100. Forest plots for all meta-analyses in this table are in online supplemental material I. These results are best interpreted alongside the information provided in online supplemental material H.
PI, prediction interval.

the prevalence of discrepancies in a representative sample across clinical disciplines, (3) the level of specificity in registrations and (4) interventions to reduce undisclosed discrepancies (see online supplemental material J for additional information about these gaps). We also identified several themes from surveying the conclusions of the articles we reviewed. These include the need for awareness surrounding discrepancies, the need for mandates, enforcement and/or new initiatives to address discrepancies, and the benefit of registering additional information such as analysis plans (online supplemental material J1 contains additional details).

## DISCUSSION

We find that outcome measures in registrations and study protocols often differ from published outcome measures, that these discrepancies are rarely disclosed, that the prevalence of discrepancies varies substantially across the articles we reviewed and that this heterogeneity is not easily assigned to specific article characteristics.

### Limitations

Given the wide range of discrepancy prevalence across individual articles, point estimates and CIs may provide false precision when extrapolating our findings to the registered literature at large. Moreover, because heterogeneity could not be explained by meta-regression of article characteristics, more precise estimates cannot be derived for subsets of the literature. The prediction intervals can reasonably be used to extrapolate to another article in the registered medical literature at large, although the included studies do not necessarily form a representative sample.

### Comparison to previous research

Our main findings are in line with those from previous systematic reviews. These reviews included 27 articles each and found that 31% of studies had a primary outcome discrepancy in the median article they reviewed[13] and 54% of studies had any outcome discrepancy in the median article they reviewed.[11] The latter review did not distinguish between primary and secondary outcomes, and many articles they reviewed only assessed primary outcomes. Our review included all the articles contained in these systematic reviews, except for a few that did not meet our inclusion criteria (eg, a PhD thesis, an abstract).

### Implications

Our review raises broader issues regarding the efficiency of the research ecosystem and the trustworthiness of research outputs. We identified articles that documented discrepancies between publications and all of registrations, protocols, ethics applications, funding applications and marketing approval applications. The existence of multiple documents outlining the same study raises the likelihood of discrepancies and, in the absence of a clearly demarcated 'master' document, leaves ambiguity

regarding which document is 'correct'. Rehashing the same study details for different audiences may also be an inefficient use of researchers' time. Identifying a single publicly accessible document as the version of record (this could be the registration) and having all other documents point to this version of record for key information could reduce ambiguity and improve efficiency.

As for trustworthiness, registration has had a clearly positive influence on medical research.[111 112] At the same time, some registration policies have poor adherence (eg, many trials are registered retrospectively, and many trial results are never reported[113]). The existence of research policies that are regularly overlooked, rarely monitored and come with no consequence for non-compliance, can be damaging in at least two ways. They risk devaluing research policies altogether and they can reduce the trustworthiness of research outputs by creating a false impression that rigorous research practices were employed. Conceiving research as a complex ecosystem comprised of various agents with diverse incentives (eg, funders, publishers, institutions, individual researchers) can help to comprehend why some policies have poor adherence and to develop and implement effective research infrastructure.

## CONCLUSION

Registrations provide the evidence to detect selective reporting and outcome switching, which we found to be common. Nearly all articles we reviewed focused on documenting issues. Future efforts regarding discrepancies—and research improvement broadly—could prove more fruitful by shifting focus towards developing and testing solutions to these now well-documented issues.

**Collaborators** Robert T Thibault (ORCID 0000-0002-6561-3962; School of Psychological Science, University of Bristol, Bristol, UK; Meta-Research Innovation Center at Stanford (METRICS), Stanford University, Stanford, California, USA); Robbie Clark (ORCID 0000-0002-2160-313X; School of Psychological Science, University of Bristol, Bristol, UK; MRC Integrative Epidemiology Unit at the University of Bristol, Bristol, UK); Hugo Pedder (ORCID 0000-0002-7813-3749; Population Health Sciences, University of Bristol, Bristol, UK; Bristol Medical School, University of Bristol, Bristol, UK); Olmo van den Akker (ORCID 0000-0002-0712-3746; Department of Methodology & Statistics, Tilburg University, Tilburg, The Netherlands; QUEST Center for Responsible Research, Berlin Institute of Health at Charité, Berlin, Germany); Samuel Westwood (ORCID 0000-0002-0107-6651; Department of Psychology, Institute of Psychiatry, Psychology, Neuroscience, King's College London, London, UK); Jacqueline M Thompson (ORCID 0000-0003-2851-3636; School of Psychological Science, University of Bristol, Bristol, UK; Bodleian Libraries, University of Oxford, Oxford, UK); Marcus R Munafò (ORCID 0000-0002-4049-993X; School of Psychological Science, University of Bristol, Bristol, UK; MRC Integrative Epidemiology Unit at the University of Bristol, Bristol, UK).

**Contributors** RTT: Conceptualisation, Data curation, Formal analysis, Funding acquisition, Investigation, Methodology, Project administration, Software, Supervision, Validation, Visualisation, Gaurantor, Writing—original draft and Writing—review and editing. RC: Investigation, Methodology and Writing—review and editing. HP: Formal analysis, Software and Writing—review and editing. OvdA: Investigation and Writing—review and editing. SW: Investigation and Writing—review and editing. JMT: Conceptualisation and Writing—review and editing. MRM: Conceptualisation, Methodology, Supervision and Writing—review and editing.

**Funding** RTT was supported by a general support grant awarded to METRICS from Arnold Ventures and postdoctoral fellowships from the Canadian Institutes of Health

Research (CIHR) and the Fonds de recherche du Québec - Santé (FRQS). HP was supported by funding from the National Institute of Health and Care Excellence's Guidelines Technical Support Unit and Bristol's National Institute for Health and Care Research Technology Assessment Group. OvdA was supported by Consolidator Grant (IMPROVE) from the European Research Council (ERC; grant no. 726361). MRM, RTT, JMT and RC were part of the MRC Integrative Epidemiology Unit (MC_UU_00011/7).

**Competing interests**  None declared.

**Patient and public involvement**  Patients and/or the public were not involved in the design, or conduct, or reporting, or dissemination plans of this research.

**Patient consent for publication**  Not applicable.

**Ethics approval**  Not applicable.

**Provenance and peer review**  Not commissioned; externally peer reviewed.

**Data availability statement**  The data are publicly available. Data, data dictionaries, analysis script and materials related to this study are publicly available on the Open Science Framework at https://osf.io/5gfty/. The study protocol and materials were registered on 17 January 2020 at https://osf.io/ktmdg. Discrepancies between this manuscript and the registered protocol are outlined in the online supplemental material A. To facilitate reproducibility, this manuscript was written by interleaving regular prose and analysis code using R Markdown. The relevant files are available in a Code Ocean container (https://doi.org/10.24433/CO.4753181.v1) which recreates the software environment in which the original analyses were performed. This container allows this manuscript to be reproduced from the data and code with a single button press.

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
