## [Reviewer comments · BMJ Open]

ARTICLE DETAILS

TITLE (PROVISIONAL)	Estimating the prevalence of discrepancies between study registrations and publications: A systematic review and meta-analyses
AUTHORS	Thibault, Robert; Clark, Robbie; Pedder, Hugo; van den Akker, Olmo; Westwood, Samuel; Thompson, Jacqueline; Munafo, Marcus

VERSION 1 – REVIEW

REVIEWER	Kurum, Esra University of California Riverside
REVIEW RETURNED	17-Oct-2021

GENERAL COMMENTS	Comments to the authors (bmjopen-2021-056372) “Estimating the prevalence of discrepancies between study registrations and publications: A systematic review and meta-analyses” This paper performs a systematic review of the quantitative data published on discrepancies between registrations or study protocols and their associated publications. The authors employed two random-effects meta-analysis approaches to analyze discrepancies in primary and secondary outcomes using data reported in 89 articles from 7000+ studies. Overall, the paper is very interesting and investigates an important issue, and contributes significant findings. I list below my suggestions/concerns regarding the paper. • The authors should include some information on the quantitative data obtained from the articles. Do they have access to raw data from these papers, or what was the format of the data obtained from these articles? Assuming that the articles used in this paper have also performed some form of meta-analysis, there is the danger of increasing the type I error rate, especially in underpowered meta-analyses that are subject to continuous updating by including more studies. There is a lot of work in this field; it has been shown via theoretical arguments as well as simulation studies and empirical analysis; I suggest the authors explore this issue in detail as it affects the validity of their results.• I suggest the authors include more information on the studies within these 89 articles; it is unclear how many studies were investigated in each article; median and range can be provided.
--

	In addition, a table similar to Table 1 can be presented in terms of studies. In addition, some examples of primary and secondary outcomes from these studies would be valuable.  • On page 9, line 8, it is mentioned that some studies were not registered at all (k=24); please clarify how did these articles evaluate the discrepancy of a study if it was not registered? • The authors mention missing data at the end of page 6; the nature of this missing data is important as it might affect the results. More detailed information on how much missing data exists should be reported. In addition, is the discrepancy information missing, or does any of the covariates used in the analysis have missing information? Authors should comment/explore whether the missing data might affect their conclusions and impose a limitation on their analysis. • Application of the random-effects meta-analysis is appropriate; however, including predictors one at a time is not appropriate. When the outcome in a regression model depends on more than one predictor, one should include all the predictors in the same model as it is known that multiple regression provides greater accuracy. In addition, this one predictor at a time approach prevents the authors from investigating the possible interactions between the predictors, which might be present and should be explored. If interaction effects are significant, the interpretations of the main effects that the authors presented would not be valid.
--	--

REVIEWER	Shi, Xiaoting Yale University School of Public Health, Department of Environmental Health Sciences
REVIEW RETURNED	28-Oct-2021

GENERAL COMMENTS	Thank you for the opportunity to review this manuscript. In this manuscript by Dr. Thibault et al., the authors conducted a systematic review to assess the discrepancies between published studies and their corresponding study protocols. The authors found selective reporting and outcome switching is common and the discrepancies cannot be simply explained by specific article characteristics, and many articles failed to provide sufficient information regarding multiple aspects including the disclosure of certain discrepancies. Overall, this is a well-designed (e.g., prespecified study protocol, and other data shared via OSF and disclosure of deviations from protocols) study, reflecting a significant amount of work (e.g., evaluation of 89 articles). To avoid ambiguity, in the Methods section, the authors clearly defined the terms used in the manuscript, and provided justification for the effect model chosen and the uniqueness of current systematic review compared to the previous reviews on similar topic. While the study contains a number of interesting findings about the discrepancies in study protocols and corresponding publications, there are several aspects where additional clarity/edits will be helpful:
---

	First, the authors did a good job by providing a definition of discrepancy in the Methods section of the manuscript; however, it remains unclear what would be considered as discrepancy and what would not. It would be helpful to give some examples that were identified from the samples (perhaps in a Box). Furthermore, the authors could even consider distinguishing major and minor discrepancies since not all discrepancies would be the same. Second, the authors specified many details of the current systematic review in the Search Strategy section in the Methods Section. In terms of the study format, it may be helpful and easier for readers to follow if the authors can have Study Selection, Eligibility Criteria as separate sections in Methods section. Additionally, many of the information in the second paragraph of Search Strategy section should be moved to Results section. Lastly, this is something the authors could consider: as a systematic review, it would be helpful to comprehensively search multiple databases or additionally manually search reference lists of the retrieved articles to make sure the thoroughness of the search. The authors searched Scopus and Web of Science, which were two widely-used medical databases. However, if possible, the authors could consider searching some additional comprehensive database(s) such as MEDLINE, or perhaps provide justification why it's not considered as searching database.
--	---

REVIEWER	Amaral, Olavo Federal University of Rio de Janeiro
REVIEW RETURNED	09-Nov-2021

GENERAL COMMENTS	I have performed this review in collaboration with Gabriel Gonçalves da Costa (https://orcid.org/0000-0002-1141-7827). The article consists of a pre-registered systematic review and meta-analysis of discrepancies between study registrations and publications. The study is generally well designed and analyzed, with data and code publicly available, although data synthesis is limited by the heterogeneity in samples and methodologies of primary studies. Perhaps because of this, a large amount of heterogeneity is observed in the results. Our comments will mostly refer to manuscript organization and to additional ways to explore this heterogeneity. Major points:  1. First of all, there are some glitches on the manuscript file. Table 2 in the submitted version appears identical to Table S1, and appears to have been erroneously submitted, as it does not correspond to Table 2 in the preprint version (which seems to match what is described in the text). Also on the topic of organization, the order of the supplementary tables makes little sense: Tables S4 and S2 come before Tables S1 and S3. Please revise. 2. The majority of the study's results are currently presented as supplementary material. Although some of these data (such as forest plots for specific discrepancies) are indeed overly detailed for the main text, I'd recommend that at least tables S1 and S2 are included in the main manuscript. Table S1 is already cited in the main text, while table S2 (non-outcome discrepancies) can probably be moved to the main manuscript, along with the text describing it, and comprise a new section on non-outcome
---

discrepancies. Table S4 (discrepancies vs. registration timing) also deserves at least to be cited in the “Parameters potentially related to discrepancies” section, although it might not have to be moved to the main text.

3. One important potential source of heterogeneity is the scientific field under study. As the vast majority of articles concern clinical research, however, “discipline” might have been coded too broadly by the authors (i.e. “medicine” contains more than 90% of the articles), preventing this from being taken into account in the analysis. Categorizing “medicine” into specific subdisciplines could be an alternative, although it could be too granular to be useful for analysis. But it would at least be interesting to stratify the analysis between general (i.e. analyzing clinical research in general) and subfield-specific primary articles, to see if heterogeneity is present in both subgroups or whether it is mostly observed among subfield-specific articles.

4. Another potential source of heterogeneity is the definition of discrepancies in the primary studies, as the authors themselves state "(e.g., whereas some articles considered a change in the timing of an outcome as a discrepancy, others did not)". The authors discuss the fact that this information was not collected, and it indeed might be hard to code into a useful taxonomy, but at least a descriptive analysis of what is normally considered a discrepancy would be welcome. This is already hinted at by the number of studies analyzing specific categories in Table S1, but could be made more visible in the main text.

Minor points:

Strengths and limitations:

- This sections mentions strengths, then a limitation, then another strength. Wouldn't it be more logical to separate strengths and limitations more clearly?
- “Spans research disciplines’ might be an overstatement, as the vast majority of articles focused on clinical trials or systematic reviews in medicine.

Introduction:

- The introduction is very focused on clinical trials, even though the abstract (and strengths and limitations) sections claim that the analysis spans research disciplines. Thus, a brief discussion of the status of registration in other areas is likely warranted.
- The mention to the Open Science Framework seems to imply that it is a repository for psychology and social sciences, but it is a general repository that has a wider span than these two disciplines.
- A scoping review that is apparently being run in parallel with this one is mentioned in the methods and protocol, and it might be worth mentioning its existence and purpose in the introduction.

Methods:

	 - There seem to be some unnecessary italics here – it’s probably enough to italicize terms in their first appearance. - The main manuscript contains no information at all on the extracted variables for each article. Although the coding information contained in Supplementary Material B is indeed to dense for the main text, the authors could consider (at their discretion) making a brief summary of the extracted variables in the main methods section. - The estimator used for the random-effects meta-analyses could be described in the methods. - The definition of a 95% confidence interval used by the author does not seem to be correct. The correct definition is that 95% of resampled meta-analyses would yield a 95% CI that contains the true value of the parameter being estimated (e.g. prevalence of discrepancies) – and not that 95% of resampled meta-analyses would give a pooled result that falls within a 95% confidence interval, as the authors state. - The definition of a 95% prediction interval could be improved by adding “on average” at some point (i.e., “95 of them, on average, to fall within the 95% prediction interval”), as not all 95% prediction intervals will include exactly 95% of samples articles. Results:  - The discussion of subfield specificity and study sampling methods in different articles (currently present in the “primary outcome discrepancies” section and in Supplementary Material I, respectively) probably should be included in the “article characteristics” section, as they are quite relevant to describe the sample of articles being analyzed. - “They sent letters to the editor within weeks of a study being published” might convey the impression that this was done for only one study in Goldacre et al. Can the authors be clearer (i.e. by providing the number of studies)? - In Table 2 (i.e. the one in the preprint, which appears to be the correct one), as well as in Tables S4, S5 and S1, it would be useful to present the point estimates for each variable along with the 95% CI and 95% PI. Discussion:  - “As for trustworthiness, registration has had a clearly positive influence on medical research.” Although I do believe this to be the case, such a strong statement should be backed up by references containing empirical evidence on the subject. Supplementary material:  - Supplementary material F: - Can’t the authors provide summarized tables rather than R outputs? This kind of formatting is not really consistent with the rest of the paper.
--	--

VERSION 1 – AUTHOR RESPONSE

Reviewer comments

Reviewer 1 summary. This paper performs a systematic review of the quantitative data published on discrepancies between registrations or study protocols and their associated publications. The authors employed two random-effects meta-analysis approaches to analyze discrepancies in primary and secondary outcomes using data reported in 89 articles from 7000+ studies. Overall, the paper is very interesting and investigates an important issue, and contributes significant findings. I list below my suggestions/concerns regarding the paper.

Comment R1.1 The authors should include some information on the quantitative data obtained from the articles. Do they have access to raw data from these papers, or what was the format of the data obtained from these articles? Assuming that the articles used in this paper have also performed some form of meta-analysis, there is the danger of increasing the type I error rate, especially in underpowered meta-analyses that are subject to continuous updating by including more studies. There is a lot of work in this field; it has been shown via theoretical arguments as well as simulation studies and empirical analysis; I suggest the authors explore this issue in detail as it affects the validity of their results.

Response R1.1 The revised manuscript clarifies information about the data we extracted:

The data we extracted was often presented as summary results in a table and sometimes as text in the results section. To be included in our meta-analyses this data had to include at least two of (1) the number of studies assessed (denominator), (2) the number of studies with a given discrepancy (numerator), and (3) the percentage of studies with a given discrepancy—form which we could calculate an unreported numerator or denominator. We did not access the raw data.

We presume it is unlikely that many of these articles share raw data (as is the case for much research: doi:10.1001/jamanetworkopen.2020.33972). Moreover, we only included studies where we could identify or calculate both the numerator and denominator. For binary data, we feel this is sufficiently close to the raw data for effective use in our meta-analysis.

The articles we obtained data from do not perform meta-analyses. Most presented the number or percentage of discrepancies as descriptive statistics. In other words, we are conducting a meta-analysis on proportions, not a meta-analysis of meta-analyses.

We agree with the reviewer's concern about Type 1 error and updating meta-analyses. However, we believe it is unlikely to be relevant for our meta-analysis. Whereas a meta-analysis that assesses the effectiveness of a treatment could impact the decision to conduct additional trials (e.g., due to clinical equipoise) and in turn lead to an increased Type 1 error rate, our meta-analysis is unlikely to impact the running of additional studies in the same way a meta-analysis of clinical trials would. Our main findings are presented as descriptive statistics and we are thus not concerned about the potential for false positives.

Comment R1.2 I suggest the authors include more information on the studies within these 89 articles; it is unclear how many studies were investigated in each article; median and range can be provided. In addition, a table similar to Table 1 can be presented in terms of studies. In addition, some examples of primary and secondary outcomes from these studies would be valuable.

Response R1.2 We thank the reviewer for this comment. To every table, we added a column that reports the number of studies (*n*) in addition to the column that already reports the number of articles (*k*). The forest plots already include both these numbers (*n* and *k*). We now also present summary statistics about the number of studies per article: Articles that checked for outcome discrepancies assessed a median of 68 studies (IQR: 33 to 112).

We have also added a box to describe discrepancies in more detail, provide concrete examples of discrepancies taken from the literature, and point readers to resources that contain detailed documentation about discrepancies in the literature.

Box 1. Examples of discrepancies.

We coded 10 types of outcome discrepancies (listed in Table 2) and 10 types of non-outcome discrepancies (listed in Table 4). The degree to which the information in a registration is discrepant with the information in a publication can range widely. The associated concern about risk of bias can also range widely and often requires domain expertise to assess. We present two examples from [54] in this box. Many more examples are available in the appendix of [54] and at <https://www.compare-trials.org/results>.

Registered: Primary outcome: Visual Acuity [Time Frame: 6 months].

Published: Primary outcome: 3-year cumulative incidence rate of myopia. Myopia was defined as a spherical equivalent refractive error (sphere +½ cylinder) of at least -0.50 D

Coded as: Timing of outcome measurement changed.

Registered: Primary outcome: Severity of Device and Procedure related complications [Time Frame: At the time of ExAblate Transcranial thalamotomy procedure]. Secondary outcome: Effectiveness of the ExAblate Transcranial MRgFUS treatment determined using the Clinical Rating Scale for Tremor (CRST) [Time Frame: Participants will be followed from the date of treatment until study completion, approximately up to 12 months].

Published: Primary outcome: Change from baseline to 3 months in the tremor score for the hand derived from the CRST, Part A (three items: resting, postural, and action or intention components of hand tremor), and the CRST, Part B (five tasks involving handwriting, drawing, and pouring).

Coded as: Secondary outcome promoted to primary outcome; Timing of outcome measurement changed; Primary outcome omitted.

Comment R1.3. On page 9, line 8, it is mentioned that some studies were not registered at all (*k*=24); please clarify how did these articles evaluate the discrepancy of a study if it was not registered?

Response R1.3 The manuscript states that articles detailed their selection process including how many studies were registered and if so, when they were registered. We now added a footnote to avoid the ambiguity that the reviewer raises: By definition, non-registered studies cannot be assessed for discrepancies between their registration and publication. Several articles identified non-registered studies in their sampling process, but these studies were not included in the final sample of articles that compared registrations to publications.

Non-registered studies may have formed part of the sample for the 16 articles that compared publications to ethics applications, other protocols, grant applications, or marketing applications.

Comment R1.4 The authors mention missing data at the end of page 6; the nature of this missing data is important as it might affect the results. More detailed information on how much missing data exists should be reported. In addition, is the discrepancy information missing, or does any of the covariates

used in the analysis have missing information? Authors should comment/explore whether the missing data might affect their conclusions and impose a limitation on their analysis.

Response R1.4 We thank the reviewer for altering us to this lack of clarity. We removed the sentence on missing data and replaced it with the following text:

If an article did not report data for a certain measure (e.g., secondary outcome discrepancies), then we did not include that article in the meta-analysis for that measure (this is why k varies among the meta-analyses we present—see Tables 2 and 4). For the meta-regressions, we could not find data on (1) the version of the registry that publications were compared to for 35 articles, and (2) the number of studies that disclosed discrepancies for 57 articles. We coded these cases as “not reported” and included “not reported” as a factor in the meta-regressions. All other meta-regression data was complete.

As presented in the text above, our meta-analyses do not analyze articles with missing data, but rather only include articles that report their results in such a way that the data can be used for the particular meta-analysis. Given that there is no standard for how to present results about discrepancies, there is a possibility that authors could selectively report (e.g., report only on primary outcomes discrepancies, but not secondary outcomes discrepancies). To partially overcome this issue, we present the results of meta-analyses on 10 types of outcome discrepancies (see Table 2). Of the 80 articles that reported on outcome discrepancies, 70 of them reported specifically on primary outcome discrepancies (which constitutes the main analysis in our manuscript). We feel this data increases confidence that articles are not selectively avoiding reporting primary outcome discrepancies.

Comment R1.5 Application of the random-effects meta-analysis is appropriate; however, including predictors one at a time is not appropriate. When the outcome in a regression model depends on more than one predictor, one should include all the predictors in the same model as it is known that multiple regression provides greater accuracy. In addition, this one predictor at a time approach prevents the authors from investigating the possible interactions between the predictors, which might be present and should be explored. If interaction effects are significant, the interpretations of the main effects that the authors presented would not be valid.

Response R1.5 In line with the reviewer’s recommendation, we now perform an additional meta-regression using all the predictors at the same time. We also conducted a sensitivity analysis that included all six regressors in a single meta-regression and found that publications compared to the most recent version of a registration ($k = 15$) may have a smaller proportion of studies with at least one primary outcome discrepancy as compared to publications compared to the original version of a registration ($k = 29$) ($p = 0.08$; OR 95% CI: 0.32 to 1.07). All meta-regression model summaries are presented in Supplementary Material E.

Supplementary Material E contains additional information:

We ran the meta-regressions presented in the manuscript with one regressor at a time. In line with a suggestion from a reviewer, we now also run a meta-regression with all regressors at the same time. We had no *a priori* thoughts on how the regressors may interact and thus did not include the interaction of regressors in our meta-regressions.

This meta-regression revealed 6 predictors at $p < 0.10$. The full output is appended below. Five of the predictors include between 1 to 4 articles in one category. These results may stem from idiosyncrasies of one or more individual studies, rather than from the predictor itself. We do not

explore these findings further as Cochrane advises against meta-regression when $k < 10$ (<https://training.cochrane.org/handbook/current/chapter-13#section-13-3-5-6>).

This meta-regression also revealed one predictor of potential interest. Articles that compared publications to the most recent version of a registration ($k = 15$) may be less likely to have at least one primary outcome discrepancy than articles that compared publications to the original version of a registration ($k = 29$) ($p = 0.08$; OR 95% CI: 0.32 to 1.07).

Supplementary Material E presents the model output for this meta-regression.

We also believe that including predictors one at a time is appropriate because it should allow identification of any large effects from individual regressors. Including all regressors simultaneously could lead to an underpowered analysis, which could prevent us from identifying if any of them are statistically important factors. There may be interaction effects, but we feel that our sample size of articles is insufficient to explore these interactions.

Reviewer 2 summary. Thank you for the opportunity to review this manuscript. In this manuscript by Dr. Thibault et al., the authors conducted a systematic review to assess the discrepancies between published studies and their corresponding study protocols. The authors found selective reporting and outcome switching is common and the discrepancies cannot be simply explained by specific article characteristics, and many articles failed to provide sufficient information regarding multiple aspects including the disclosure of certain discrepancies. Overall, this is a well-designed (e.g., prespecified study protocol, and other data shared via OSF and disclosure of deviations from protocols) study, reflecting a significant amount of work (e.g., evaluation of 89 articles). To avoid ambiguity, in the Methods section, the authors clearly defined the terms used in the manuscript, and provided justification for the effect model chosen and the uniqueness of current systematic review compared to the previous reviews on similar topic. While the study contains a number of interesting findings about the discrepancies in study protocols and corresponding publications, there are several aspects where additional clarity/edits will be helpful:

Comment R2.1 First, the authors did a good job by providing a definition of discrepancy in the Methods section of the manuscript; however, it remains unclear what would be considered as discrepancy and what would not. It would be helpful to give some examples that were identified from the samples (perhaps in a Box). Furthermore, the authors could even consider distinguishing major and minor discrepancies since not all discrepancies would be the same.

Response R2.1 We thank the reviewer for this suggestion and have added a box with examples of discrepancies (**Response 1.2** contains additional detail on this comment).

We collected data that distinguished 10 types of outcome discrepancies (e.g., primary outcome omitted, secondary outcome added). This information was previously in the supplementary material. We have now moved it to the main manuscript in Table 2. We also moved the same table on non-outcome discrepancies to the main manuscript as Table 4.

We are not able to identify whether discrepancies within each of these 10 categories are major or minor. The main reason being that the articles we reviewed do not make this major/minor distinction. Moreover, they do not provide descriptions of each discrepancy—which would be needed to make this judgment ourselves. In this box, we also highlight that domain expertise is often needed to make judgments about the extent to which a discrepancy raises concerns about bias. Given all these considerations, we refrained from making these judgments.

Comment R2.2 Second, the authors specified many details of the current systematic review in the Search Strategy section in the Methods Section. In terms of the study format, it may be helpful and easier for readers to follow if the authors can have Study Selection, Eligibility Criteria as separate sections in Methods section. Additionally, many of the information in the second paragraph of Search Strategy section should be moved to Results section.

Response R2.2 In line with the reviewer's recommendation, we have split this section of the text into "Eligibility criteria" and "Study selection".

We have chosen to keep information about inter-rater agreement in the methods section. We do this in order to keep details about the study selection in one spot and near to the PRISMA flowchart. In particular, many of the articles we include (33 of the 89) come from a snowball method. We maintain the chronological order of the study selection and added a sentence to emphasize that the inter-rater agreement was based on our search before employing the snowball method:

...Inter-rater agreement for all 4,283 articles was Cohen's $k = 0.72$. We then used a snowball method and identified 33 additional articles that met our inclusion criteria, mostly through citations in [7] and [9]. These 33 articles are not included in the inter-rater agreement scores. After full-text review, we included 89 articles in our systematic review.

Comment R2.3 Lastly, this is something the authors could consider: as a systematic review, it would be helpful to comprehensively search multiple databases or additionally manually search reference lists of the retrieved articles to make sure the thoroughness of the search. The authors searched Scopus and Web of Science, which were two widely-used medical databases. However, if possible, the authors could consider searching some additional comprehensive database(s) such as MEDLINE, or perhaps provide justification why it's not considered as searching database.

Response R2.3 In line with the reviewer's suggestion, we had already manually searched the reference lists of the three systematic reviews we found on this topic:

We then used a snowball method and identified 33 additional articles that met our inclusion criteria, mostly through citations in [7] and [9].

We agree that extending the search query to include additional databases and manually searching the reference list of each individual article may capture more articles. However, based on the information provided in **Response E4**, we presume that including additional articles will not substantially impact our conclusions and we do not have the resources to complete this task.

Reviewer 3 summary. The article consists of a pre-registered systematic review and meta-analysis of discrepancies between study registrations and publications. The study is generally well designed and analyzed, with data and code publicly available, although data synthesis is limited by the heterogeneity in samples and methodologies of primary studies. Perhaps because of this, a large amount of heterogeneity is observed in the results. Our comments will mostly refer to manuscript organization and to additional ways to explore this heterogeneity.

Comment R3.1 First of all, there are some glitches on the manuscript file. Table 2 in the submitted version appears identical to Table S1, and appears to have been erroneously submitted, as it does not correspond to Table 2 in the preprint version (which seems to match what is described in the text). Also

on the topic of organization, the order of the supplementary tables makes little sense: Tables S4 and S2 come before Tables S1 and S3. Please revise.

Response R3.1 We thank the reviewer for bringing this issue to our attention. We have replaced Table 2 in the manuscript file with the correct table. As per **Response R3.2**, we have also moved Table S1 and S2 to the manuscript and renumbered the tables accordingly.

Comment R3.2 The majority of the study's results are currently presented as supplementary material. Although some of these data (such as forest plots for specific discrepancies) are indeed overly detailed for the main text, I'd recommend that at least tables S1 and S2 are included in the main manuscript. Table S1 is already cited in the main text, while table S2 (non-outcome discrepancies) can probably be moved to the main manuscript, along with the text describing it, and comprise a new section on non-outcome discrepancies. Table S4 (discrepancies vs. registration timing) also deserves at least to be cited in the "Parameters potentially related to discrepancies" section, although it might not have to be moved to the main text.

Response R3.2 As per the reviewer's recommendations, we have now:

- Moved Table S1 to the manuscript (now Table 2).
- Moved Table S2 plus the associated paragraph to the manuscript in a new subsection of the results entitled "Non-outcome discrepancies" (now Table 4).
- Include a reference to Supplementary Material D, which begins with Table S4 (now Supplementary Table D1)

Comment R3.3 One important potential source of heterogeneity is the scientific field under study. As the vast majority of articles concern clinical research, however, "discipline" might have been coded too broadly by the authors (i.e. "medicine" contains more than 90% of the articles), preventing this from being taken into account in the analysis. Categorizing "medicine" into specific subdisciplines could be an alternative, although it could be too granular to be useful for analysis. But it would at least be interesting to stratify the analysis between general (i.e. analyzing clinical research in general) and subfield-specific primary articles, to see if heterogeneity is present in both subgroups or whether it is mostly observed among subfield-specific articles.

Response R3.3 We agree with the reviewer that specific sub-disciplines may have more discrepancies than others. We coded discipline broadly based on the 26 research categories used by Scopus. We were aware that these disciplines are broad, and thus extracted additional information in the dataset variable 'disciplineDetail'. We have now visually explored this variable to consider the potential of classifying articles in the discipline "Medicine" into sub-disciplines. However, we only find a few articles in each clearly demarcated sub-discipline. For example, about five articles focus specifically on oncology. However, three of these articles focus on oncology in general, one focuses on pharmaceutical interventions, and one focuses on radiation treatment. Moreover, a few articles not categorized as oncology do assess oncology studies, but also studies in other disciplines—and these results are collapsed together in our dataset. Each of these articles likely has several other idiosyncrasies which could lead to differences in results.. As such, the number of studies within the sub-disciplines do not meet the minimum of 10 studies that Cochrane suggests (as discussed in **Response 1.5**); the sub-disciplines we can identify are too granular to justify additional meta-regressions, as was also mentioned by the reviewer.

We have added the following footnote: If readers want to find if a study on discrepancies exists within a particular sub-discipline, we invite them to explore the variable 'disciplineDetail' in our dataset.

Comment R3.4 Another potential source of heterogeneity is the definition of discrepancies in the primary studies, as the authors themselves state "(e.g., whereas some articles considered a change in the timing of an outcome as a discrepancy, others did not)". The authors discuss the fact that this information was not collected, and it indeed might be hard to code into a useful taxonomy, but at least a descriptive analysis of what is normally considered a discrepancy would be welcome. This is already hinted at by the number of studies analyzing specific categories in Table S1, but could be made more visible in the main text.

Response R3.4 In line with this comment and others from the reviewer, we have moved Table S1 to the main manuscript. This table outlines the rate of discrepancies for 10 different types of outcome discrepancies (e.g., changing a primary outcome to a secondary outcome; a change in the timing of an outcome assessment). We have also added Box 1 (see **Response 1.2**), which presents a few examples of discrepancies from the articles we analyzed.

The reviewer is correct that the definition in the meta-analysis for "Any primary outcome discrepancy" contains articles that use different definitions of primary outcomes. With this in mind, we also ran meta-analyses as explained in the following manuscript excerpt, which deals with this issue:

We did not collect information on the exact definitions an article used to identify a primary outcome discrepancy. However, we did collect information on the proportion of articles with sub-categories of outcome discrepancies, which are more strictly defined and listed in Table 2 (e.g., promoting a secondary outcome to a primary outcome). We ran meta-analyses on these sub-categories of outcome discrepancies and found they also had high heterogeneity (see Table 2). Thus, varying definitions are unlikely to be the main driver of the high heterogeneity in the present analysis on primary outcome discrepancies.

Minor comments

Comment R3.5 Strengths and limitations: This sections mentions strengths, then a limitation, then another strength. Wouldn't it be more logical to separate strengths and limitations more clearly?

Response R3.5 We thank the reviewer for the comment and have rearranged this section and added another limitation. We now present three strengths followed by two limitations. We checked a few other BMJ Open articles and this format of strengths then weaknesses appears common. Given that the BMJ Open Author Guidelines ask for a single section entitled "Strengths and limitations of this study", we cannot separate the strengths and limitations under separate headers.

- We employ a wide-reaching search strategy and captured 89 articles including over 6,000 registrations and publications.
- Our coding procedure includes fine-grained information that allowed us to run meta-regressions and test whether several parameters impact discrepancies.
- All our data and code are openly available.
- The high heterogeneity in the meta-analyses led to wide-ranging confidence intervals and prediction intervals.
- Many articles did not fully operationalize their definition of what constitutes a discrepancy (e.g., which version of the registration was used).

Comment R3.6 "Spans research disciplines" might be an overstatement, as the vast majority of articles focused on clinical trials or systematic reviews in medicine.

Response R3.6 We understand how this sentence could be understood in a different way than we intended. Although almost all articles we identified were in medicine, we did “employ a wide-reaching search strategy that spans research disciplines”, in that we used search terms from medicine, psychology, economics, and other fields (e.g., prospective registration, preregistration, pre-analysis plans). It just happens that there were few relevant articles in most disciplines other than medicine. We have reworded the bullet point to avoid ambiguity:

- We employ a wide-reaching search strategy and captured 89 articles including over 6,000 registrations and publications.

Comment R3.7 The introduction is very focused on clinical trials, even though the abstract (and strengths and limitations) sections claim that the analysis spans research disciplines. Thus, a brief discussion of the status of registration in other areas is likely warranted.

Response R3.7 We thank the reviewer for this comment and have added a paragraph to the introduction:

In research disciplines other than clinical trials, study registration is becoming more common, but remains far from standard practice [e.g., 109-111]. For example, since about 2011 the field of psychology has increasingly taken the “replication crisis” seriously and many researchers and journals now use registration to reduce bias and make risk of bias transparent [119]. Other disciplines have created dedicated registries, such as PROSPERO for systematic reviews and the American Economic Association RCT Trial Registry.

Comment R3.8 The mention to the Open Science Framework seems to imply that it is a repository for psychology and social sciences, but it is a general repository that has a wider span than these two disciplines.

Response R3.8 We thank the reviewer for this comment and have edited the sentence to read:

...registration has expanded beyond clinical trials; we included all research disciplines and used key word searches for registries including the Open Science Framework, the American Economic Association RCT Trial Registry, and PROSPERO.

Comment R3.9 A scoping review that is apparently being run in parallel with this one is mentioned in the methods and protocol, and it might be worth mentioning its existence and purpose in the introduction.

Response R3.9 This scoping review has since been terminated because the student leading it changed career paths and reviews with similar goals have been published (e.g., Hardwicke & Wagenmakers 2023, DeVito 2022.) We have updated the preregistration (<https://osf.io/ktmdg>) to highlight that the scoping review was terminated and now outline this information in a manuscript footnote:

We had planned to report on this scoping review separately. However, the scoping review has been terminated (details provided at <https://osf.io/ktmdg>).

This link points to an updated version of our registration that states:

This update was made on May 19, 2023

Reason for update:

This preregistration outlined plans for (1) a scoping review, and (2) a systematic review. For two reasons, the scoping review was put on pause in July 2020 and terminated in May 2023. First, the student leading the scoping review changed career paths. Second, before we assigned someone else to lead the scoping review to completion, at least two papers covering similar topics were published.

Hardwicke, T. E., & Wagenmakers, E. J. (2023). Reducing bias, increasing transparency and calibrating confidence with preregistration. *Nature Human Behaviour*, 7(1), 15-26.

De Vito, N. J. (2022). Trial registries for transparency and accountability in clinical research (Doctoral dissertation, University of Oxford).

At the time of termination, we had performed three tasks. (1) Completed an initial literature search. The list of relevant papers we identified is available in the dataset published with the Systematic Review (<https://doi.org/10.1101/2021.07.07.21259868>). (2) We created a Qualtrics form to extract data (available at <https://osf.io/2pfea>). (3) One researcher (RTT) completed the Qualtrics form for 64 papers. The extracted data is available at <https://osf.io/xvn7p>. Note, this data is incomplete—it only includes some of the relevant papers identified and has not been cleaned. We are sharing this data to maximize transparency about what state the project was in when it was terminated.

Comment R3.10 Methods: There seem to be some unnecessary italics here – it's probably enough to italicize terms in their first appearance.

Response R3.10 We only italicize these words in the “Terminology” section of the methods. All words are only italicized once, except for studies and articles, which are italicized twice. We feel these italics in this sentence will facilitate the reader’s understanding.

Comment R3.11 The main manuscript contains no information at all on the extracted variables for each article. Although the coding information contained in Supplementary Material B is indeed to dense for the main text, the authors could consider (at their discretion) making a brief summary of the extracted variables in the main methods section.

Response R3.11 We added a sentence to the “Coding items” section of the methods to address this comment:

The form consisted of five sections that assessed (1) article characteristics, (2) study registration details, (3) 10 types of outcome discrepancies (listed in Table 2), (4) 10 types of non-outcome discrepancies (listed in Table 4), and (5) any additional descriptive or inferential statistics on discrepancies.

We also added information to the “Coding items” section as per **Response 1.1** and **Response 1.4**.

Comment R3.12 The estimator used for the random-effects meta-analyses could be described in the methods.

Response R3.12 The manuscript now addresses this point:

We used a random intercept logistic regression model with the Knapp-Hartung adjustment for the synthesis of proportions [113]. We used the maximum-likelihood method for estimating the between-study heterogeneity (Tau).

Comment R3.13 The definition of a 95% confidence interval used by the author does not seem to be correct. The correct definition is that 95% of resampled meta-analyses would yield a 95% CI that contains the true value of the parameter being estimated (e.g. prevalence of discrepancies) – and not that 95% of resampled meta-analyses would give a pooled result that falls within a 95% confidence interval, as the authors state.

Response R3.13 We thank the reviewer for this correction. The text now reads:

If we assume that we could resample from our population, 95% of the resampled *meta-analyses* would yield a 95% confidence interval that contains the true value of the parameter being estimated (e.g., prevalence of discrepancies).

Comment R3.14 The definition of a 95% prediction interval could be improved by adding “on average” at some point (i.e., “95 of them, on average, to fall within the 95% prediction interval”), as not all 95% prediction intervals will include exactly 95% of samples articles.

Response R3.14 We agree with the reviewers suggestion. The text now reads:

Alternatively, if we are interested in the results that would come from another article assessing discrepancies, we would want a 95% prediction interval. In other words, of 100 articles drawn from the same population, we could expect the results from 95 of them—on average—to fall within the 95% prediction interval.

Comment R3.15 Results: The discussion of subfield specificity and study sampling methods in different articles (currently present in the “primary outcome discrepancies” section and in Supplementary Material I, respectively) probably should be included in the “article characteristics” section, as they are quite relevant to describe the sample of articles being analyzed.

Response R3.15 Much of this information is included in Table 1, which is presented in the *Article Characteristics* section. We agree that some of the information could appear in the Article Characteristics section. However, we feel that this will hamper the flow of the article. This information is very important for interpreting the meta-regression results and thus, I think it is appropriate for it to appear in the meta-regression paragraph of the Primary Outcome Discrepancies section.

Comment R3.16 “They sent letters to the editor within weeks of a study being published” might convey the impression that this was done for only one study in Goldacre et al. Can the authors be clearer (i.e. by providing the number of studies)?

Response R3.16 We have edited this text to read:

Only one article attempted to correct published discrepancies [16]. The authors assessed all trials published in five journals over a six-week period and sent a letter to the editor for each trial that published a discrepant outcome (for 58 letters in total).

Comment R3.17 In Table 2 (i.e. the one in the preprint, which appears to be the correct one), as well as in Tables S4, S5 and S1, it would be useful to present the point estimates for each variable along with the 95% CI and 95% PI.

Response R3.17 Many of our results have wide confidence intervals and very wide prediction intervals. With this in mind, we decided to not include point estimates because we presume that those who read or cite our results will often focus on the point estimates. We feel this would be an incomplete representation of our results. We understand that this reporting method is non-standard, but feel it is justified.

Comment R3.18 Discussion: “As for trustworthiness, registration has had a clearly positive influence on medical research.” Although I do believe this to be the case, such a strong statement should be backed up by references containing empirical evidence on the subject.

Response R3.18 We thank the reviewer for this comment and have added references to a specific example of the benefits of preregistration (Kaplan & Irvin, 2015) and a thorough treatise on registration in clinical trials (deVito 2022).

Comment R3.19 Supplementary material F: Can't the authors provide summarized tables rather than R outputs? This kind of formatting is not really consistent with the rest of the paper.

Response R3.19 We present information about the meta-regressions in two places. First, we present high level summary information in the results section of the manuscript. This includes the p-values and confidence intervals that we feel a reader can quickly digest to understand the main results. Second we include the complete summary information as R output in the supplementary material. We do not expect many readers to consult this supplementary material. However, for those who do, we want this information to be thorough and reproducible. We feel that presenting R output achieves both these criteria more effectively than a summary table or another format would. For example, standard packages for presenting metaregression results (such as the *texreg* package remove information we deem important to include in the supplementary material).

Additional notes from the authors. While revising the manuscript to address the recommendations of the editor and reviewers, we (the authors) noticed a few other improvements that could be made to the manuscript. We outline the additional changes we made below.

A1. We previously reported prediction intervals before confidence intervals. We now report confidence intervals first and prediction intervals second. We feel that readers will be more familiar with confidence intervals and that they are meaningful summary results for our meta-analysis.

A2. We provide a clarification in Supplementary Material F: In our initial coding form, we identified 19 articles reporting that the studies they assessed disclosed one or more discrepancies and 9 articles reporting that the studies they assessed disclosed no discrepancies (28 articles total). Only 21 of these 28 articles reported the number of studies assessed and the number of discrepancies disclosed (or percentages from which we could calculate these values). This forest plot only includes data from those 21 articles.

A3. We added the following text to the manuscript: In the time since our literature search, at least two interventional studies were published [114, 115]. One reports a trial that attempted to reduce

discrepancies at medical journals by sending peer reviewers information about the study registration [114]. They found null results. The other was a feasibility study that assigned a peer reviewer to specifically check for discrepancies in manuscripts submitted for publication [115].

Thank you again for the thoughtful comments which we feel have strengthened our manuscript.

Robert Thibault, on behalf of all the authors.

VERSION 2 – REVIEW

REVIEWER	Shi, Xiaoting Yale University School of Public Health, Department of Environmental Health Sciences
REVIEW RETURNED	06-Jun-2023
GENERAL COMMENTS	Thank you for the opportunity to continue reviewing this manuscript. The authors have responded and made changes to most of my comments. To make the manuscript even better, my last nit-picky point would be to thoroughly check the manuscript and make sure the correct tense is consistently used (for example some parts in the methods section and the last paragraph of the introduction should be written in past tense).

VERSION 2 – AUTHOR RESPONSE

In response to the sole reviewer comment on our first resubmission, we have changed several verbs to the past tense. This is viewable in the tracked changes version of our current resubmission

In our first resubmission, we added a few references. We have now ensured that these references are numbered in the order they appear in the text. We have also corrected a few minor typos, uploaded our data and code to a repository, and updated our data availability statement accordingly. These changes are also viewable in the tracked changes version.

We are looking forward to receiving the proofs. Thank you.

Kind regards,
Robert Thibault